# HypCBC: Domain-Invariant Hyperbolic Cross-Branch Consistency for Generalizable Medical Image Analysis

**Francesco Di Salvo**                                    *francesco.di-salvo@uni-bamberg.de*
*xAILab Bamberg*
*University of Bamberg, Germany*

**Sebastian Doerrich**                                    *sebastian.doerrich@uni-bamberg.de*
*xAILab Bamberg*
*University of Bamberg, Germany*

**Jonas Alle**                                    *jonas.alle@uni-bamberg.de*
*xAILab Bamberg*
*University of Bamberg, Germany*

**Christian Ledig**                                    *christian.ledig@uni-bamberg.de*
*xAILab Bamberg*
*University of Bamberg, Germany*

**Reviewed on OpenReview:** *https://openreview.net/forum?id=1spGpYmDjy*

## Abstract

Robust generalization beyond training distributions remains a critical challenge for deep neural networks. This is especially pronounced in medical image analysis, where data is often scarce and covariate shifts arise from different hardware devices, imaging protocols, and heterogeneous patient populations. These factors collectively hinder reliable performance and slow down clinical adoption. Despite recent progress, existing learning paradigms primarily rely on the Euclidean manifold, whose flat geometry fails to capture the complex, hierarchical structures present in clinical data. In this work, we exploit the advantages of hyperbolic manifolds to model complex data characteristics. We present the first comprehensive validation of hyperbolic representation learning for medical image analysis and demonstrate statistically significant gains across eleven in-distribution datasets and three ViT models. We further propose an unsupervised, domain-invariant hyperbolic cross-branch consistency constraint. Extensive experiments confirm that our proposed method promotes domain-invariant features and outperforms state-of-the-art Euclidean methods by an average of $+2.1\%$ AUC on three domain generalization benchmarks: Fitzpatrick17k, Camelyon17-WILDS, and a cross-dataset setup for retinal imaging. These datasets span different imaging modalities, data sizes, and label granularities, confirming generalization capabilities across substantially different conditions. The code is available at github.com/francescodisalvo05/hyperbolic-cross-branch-consistency.

## 1 Introduction

Deep learning models have achieved remarkable success over the past decade, yielding exceptional results in various computer vision tasks ranging from image classification to instance segmentation. However, ensuring that models generalize reliably across distribution shifts remains a fundamental challenge, especially in safety-critical applications. For instance, autonomous driving systems must generalize across varying scenery, lighting, and weather conditions at test time (Bijelic et al., 2020; Kumar & Muhammad, 2023). Similarly, medical imaging models face different domain generalization challenges, *e.g.*, shifts in patient populations,

tissue-staining protocols, or scanner manufacturers (Ktena et al., 2024; Čevora et al., 2024). These may seem subtle compared to outdoor settings, yet they still strongly affect performance. Robustness to such subtle but consequential domain changes is therefore essential for safe deployment of AI in hospitals and clinics. Most domain-shift remedies in medical imaging span data augmentations (Zhang et al., 2018; Di Salvo et al., 2024) and Euclidean representation learning (Arjovsky et al., 2019; Sagawa et al., 2020; Krueger et al., 2021).

While effective in many settings, Euclidean embeddings offer a uniform, flat geometry that may not align with the hierarchical relationships often present in clinical data. On the other hand, the hyperbolic manifold has gained notable traction in recent years (Mettes et al., 2024). Indeed, it offers a natural remedy: its constant negative curvature mirrors hierarchical structures by allocating exponentially increasing space for finer-grained distinctions, and it has shown notable performance in many vision tasks. However, end-to-end hyperbolic networks can be unstable on large datasets (Ayubcha et al., 2024) and remain underutilized in medical imaging. To overcome these stability challenges, we project Euclidean embeddings from a frozen foundation model into a lightweight hyperbolic manifold and demonstrate its clear advantages over Euclidean baselines across eleven medical datasets.

We then introduce HypCBC, a hyperbolic two-branch training strategy with domain-invariant cross-branch consistency (CBC) regularization, as illustrated in Figure 1. This approach learns both fine-grained and domain-agnostic representations, yielding consistent and substantial domain generalization improvements on three datasets: Fitzpatrick17k (dermatology), Camelyon17-WILDS (histopathology), and a cross-dataset retinal imaging benchmark.

To motivate and validate the learning dynamics of our two-branch strategy, we also report two targeted ablation studies. First, we vary the latent dimensionality of a single-branch model to quantify how its capacity influences domain invariance versus label discrimination. Second, we compare low-dimensional against high-dimensional regularization, demonstrating that the former yields significant robustness gains that cannot be explained by a mere increase of parameter count. In summary, our contributions are:

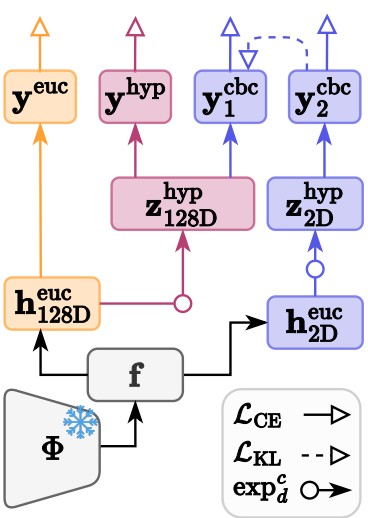

- We demonstrate that hyperbolic embeddings significantly outperform ($p < 0.05$) Euclidean ones in classification accuracy across eleven in-distribution (ID) medical imaging datasets. These span a diverse range of imaging modalities (9), sample sizes ($10^2-10^5$), and label granularities ($2-11$).

- We introduce a novel hyperbolic two-branch training strategy with a domain-invariant consistency constraint. This approach significantly enhances domain generalization (DG) performance, measured by the area under the receiver operating curve (AUC), in dermatology, histopathology, and retinal imaging.

- Ablation studies on bottleneck dimension and manifold geometry reveal two key findings. First, a 2D low-dimensional branch effectively balances domain invariance with label discrimination. Second, hyperbolic regularization exhibits consistent robustness improvements, unlike its Euclidean counterpart.

Figure 1: Given a frozen Euclidean feature extractor $\Phi$ that outputs $\mathbf{f}$, ERM applies a Euclidean linear probe over a 128D projection $\mathbf{h}_{128D}$. HypERM additionally uses a fixed exponential map $\exp_{128}^c$ to classify over hyperbolic embeddings $\mathbf{z}_{128D}$. Our method, HypCBC, introduces a second projection $\mathbf{h}_{2D}$ followed by $\exp_2^c$ to yield $\mathbf{z}_{2D}$. The logits of this low-dimensional branch are used as targets in the KL loss, promoting domain-agnostic information transfer into the high-dimensional branch.

# 2 Related work

## 2.1 Hyperbolic manifold

Hyperbolic spaces have emerged as a powerful tool for modeling hierarchical and tree-like data structures (Mettes et al., 2024). Recent work demonstrates their effectiveness across natural language processing (Dhingra et al., 2018), few-shot learning (Guo et al., 2022; Khrulkov et al., 2020), hierarchical classification (Dhall et al., 2020), metric learning (Ermolov et al., 2022; Bi et al., 2025), semantic segmentation (Atigh et al., 2022), out-of-distribution detection (Guo et al., 2022), category discovery (Liu et al., 2025), and anomaly detection (Li et al., 2024; Gonzalez-Jimenez et al., 2025). Despite these successes, hyperbolic representations remain underexplored in medical contexts. Existing efforts include fine-grained classification (Yu et al., 2022; Ramirez et al., 2025), multi-modal neuroimaging (Ayubcha et al., 2024), and anomaly detection (Gonzalez-Jimenez et al., 2025). However, end-to-end hyperbolic networks can exhibit numerical instability and substantially reduced training efficiency, with convergence requiring significantly more epochs, an effect that becomes more pronounced on larger datasets (Ayubcha et al., 2024). To combine the strengths of hyperbolic embeddings with stable training, we follow the projection-based approach of Ermolov et al. (2022), but unlike their end-to-end fine-tuning of the backbone, we freeze a pre-trained Euclidean backbone and append lightweight hyperbolic projection layers. We evaluate these representations for domain generalization, analyzing how embedding dimensionality influences the separation of domain-specific versus domain-agnostic features.

## 2.2 Domain Generalization

The community has developed a variety of domain generalization methods, primarily representation-learning techniques such as adversarial learning (Ganin et al., 2016), invariant risk minimization (Arjovsky et al., 2019), and meta-learning (Li et al., 2018a), often relying on domain labels. To boost robustness, image and latent augmentation strategies such as AugMix (Hendrycks et al., 2019) and MixStyle (Zhou et al., 2024) are also commonly utilized. Furthermore, recent works have shown that targeted augmentations offer greater gains than domain-agnostic ones (Gao et al., 2023; Di Salvo et al., 2024).

Our approach takes advantage of the hyperbolic manifold, whose constant negative curvature more accurately reflects the complexity of clinical data. Similar to Domain Adversarial Neural Networks (DANN) (Ganin et al., 2016), we introduce a second branch to promote domain invariance throughout the network. However, instead of relying on domain labels and an adversarial training strategy, our method achieves invariance in a fully unsupervised manner. We accomplish this goal by employing a low-dimensional manifold, which has been empirically demonstrated not to discriminate between domains (*cf.* Section 5.1).

Concurrently, Bi et al. (2025) embed hyperbolic geometry directly into VMamba via end-to-end state-space modeling for fine-grained domain generalization. By contrast, our method is backbone-agnostic and introduces lightweight hyperbolic projections with cross-branch consistency on frozen features, targeting unsupervised domain generalization with minimal architectural overhead.

## 2.3 Information transfer and consistency

Knowledge distillation, originally proposed to compress large "teacher" networks into smaller "students", has also been adopted for domain generalization as a form of regularization that transfers domain-invariant information. Empirical studies suggest that early network layers often encode domain-specific information (Zhou et al., 2024). To this extent, prior works exploit this property by distilling final-layer predictions into randomly selected intermediate classifiers to encourage invariance throughout the network (Sultana et al., 2022). Subsequent refinements introduce logit softening to further stabilize the distillation process (Galappaththige et al., 2024).

In this work, we reinterpret such knowledge transfer as a *cross-branch consistency regularization*. Specifically, the low-dimensional (*i.e.*, domain-invariant) hyperbolic branch provides a compact reference that constrains the high-dimensional branch via a consistency objective, encouraging domain-invariant representations without relying on classical teacher-student distillation.

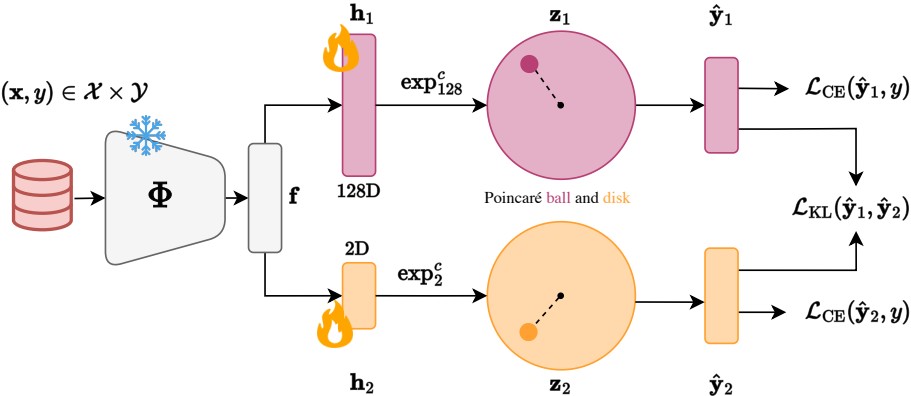

Figure 2: Given an input image-label pair $(\mathbf{x}, y) \in \mathcal{X} \times \mathcal{Y}$, we extract an image embedding $\mathbf{f} = \Phi(\mathbf{x}) \in \mathbb{R}^n$, where $n$ depends on the chosen backbone. This is projected via two heads into Euclidean embeddings $\mathbf{h}_1 \in \mathbb{R}^{128}$ and $\mathbf{h}_2 \in \mathbb{R}^2$. Each is mapped into its respective Poincaré ball $\mathbb{D}_c^d$ by $\exp_d^c$, yielding hyperbolic embeddings $\mathbf{z}_1$ and $\mathbf{z}_2$. Both branches incur cross-entropy losses on their Multiclass Logistic Regression (MLR) logits $\hat{\mathbf{y}}_1$ and $\hat{\mathbf{y}}_2$. In addition, $\hat{\mathbf{y}}_2$ also supervises $\hat{\mathbf{y}}_1$ via a KL-based consistency term. The high-dimensional branch captures fine-grained, class-specific features for inference, and the low-dimensional branch enforces an information bottleneck that promotes domain-invariant representations.

While Euclidean knowledge transfer (*e.g.*, distillation) is well studied in prior works, to the best of our knowledge, its application in hyperbolic space remains limited. Recent work by Yang et al. (2025) introduce a hyperbolic knowledge distillation approach for cross-domain few-shot learning. Their method relies on multiple domain-specific teacher models, meta-learning, and access to target-domain data at test time, placing it closer to domain adaptation than domain generalization. In contrast, our setting assumes no domain labels and no access to target-domain data at any stage. We thus focus on improving robustness to entirely unseen domains under a standard domain generalization protocol.

## 3 Method

### 3.1 Hyperbolic space

The $n$-dimensional hyperbolic space $\mathbb{H}^n$ naturally offers a geometry suited for complex, hierarchical data structures, such as medical images. While Euclidean space has flat geometry, *i.e.*, with curvature $c = 0$, hyperbolic space has a *constant negative curvature*, which can effectively capture the inherent hierarchical feature relations of image data (Mettes et al., 2024).

Among the several isometric models of hyperbolic space, we use the widely adopted Poincaré ball model $(\mathbb{D}_c^n, g^{\mathbb{D}})$ to represent an $n$-dimensional hyperbolic space (Khrulkov et al., 2020; Atigh et al., 2022; Ermolov et al., 2022; Guo et al., 2022).

Following the notation of Khrulkov et al. (2020), we define the manifold as $\mathbb{D}_c^n = \{\mathbf{x} \in \mathbb{R}^n : c\|\mathbf{x}\|^2 < 1\}$, where $c > 0$ is a scaling factor controlling the magnitude of the negative curvature. The metric (*i.e.*, rule for measuring distances) of this space is given by:

$$g^{\mathbb{D}}(\mathbf{x}) = (\lambda_{\mathbf{x}}^c)^2 g^E \quad \lambda_{\mathbf{x}}^c = \frac{2}{1 - c\|\mathbf{x}\|^2} \tag{1}$$

where $g^E = I_n$ is the standard Euclidean metric. In simple terms, the conformal factor $\lambda_{\mathbf{x}}^c$ scales the usual Euclidean distances, adapting them to the curved geometry of the Poincaré ball. To perform vector operations similar to addition in Euclidean space, the gyrovector formalism (Ungar, 2009) is adopted, which defines the Möbius addition of two points $\mathbf{x}, \mathbf{y} \in \mathbb{D}_c^n$ in Equation 2.

$$\mathbf{x} \oplus_c \mathbf{y} = \frac{(1 + 2c\langle \mathbf{x}, \mathbf{y}\rangle + c\|\mathbf{y}\|^2)\mathbf{x} + (1 - c\|\mathbf{x}\|^2)\mathbf{y}}{1 + 2c\langle \mathbf{x}, \mathbf{y}\rangle + c^2\|\mathbf{x}\|^2\|\mathbf{y}\|^2} \tag{2}$$

This operation generalizes the familiar concept of vector addition to our curved space. Based on Möbius addition, the geodesic (*i.e.*, shortest-path) distance between two points $\mathbf{x}, \mathbf{y} \in \mathbb{D}_c^n$ is given by:

$$D_{\text{hyp}}(\mathbf{x}, \mathbf{y}) = \frac{2}{\sqrt{c}} \operatorname{arctanh}\left(\sqrt{c}\| - \mathbf{x} \oplus_c \mathbf{y}\|\right) \tag{3}$$

Notably, as the curvature parameter $c$ approaches 0, the hyperbolic distance converges to twice the Euclidean distance, *i.e.*, $\lim_{c\to 0} D_{\text{hyp}}(\mathbf{x}, \mathbf{y}) = 2\|\mathbf{x} - \mathbf{y}\|$.

To fill the gap between conventional feature extractors (which operate in Euclidean space) and our hyperbolic representation, we employ the *exponential map*. This bijective mapping projects a Euclidean vector $\mathbf{v} \in \mathbb{R}^n$ onto the hyperbolic manifold at a chosen base point $\mathbf{x}_B$ (usually set to $\mathbf{0}$). The exponential map is defined as:

$$\exp_{\mathbf{x}_B}^c(\mathbf{v}) = \mathbf{x}_B \oplus_c \left(\tanh\left(\frac{\sqrt{c}\lambda_{\mathbf{x}}^c\|\mathbf{v}\|}{2}\right)\frac{\mathbf{v}}{\sqrt{c}\|\mathbf{v}\|}\right) \tag{4}$$

This exponential mapping ensures that the features are faithfully transferred from the Euclidean to the hyperbolic manifold. Its inverse, *i.e.*, the logarithmic map, allows points in the hyperbolic space to be projected back into the Euclidean space.

## 3.2 Multi-branch learning

To further exploit hyperbolic geometry for domain generalization, we draw inspiration from Domain-Adversarial Neural Networks (DANN) (Ganin et al., 2016) while eliminating the need for explicit domain labels and gradient reversal. As illustrated in Figure 2, given an image–label pair $(\mathbf{x}, y) \in \mathcal{X} \times \mathcal{Y}$, and a frozen Euclidean backbone $\Phi$, we first extract an image embedding $\mathbf{f} = \Phi(\mathbf{x}) \in \mathbb{R}^n$, where $n$ depends on the size of the chosen backbone. For "small", "base", and "large" ViT backbones, $n$ is 384, 768, and 1024, respectively.

Two projection heads $h_i : \mathbb{R}^n \to \mathbb{R}^{d_i}$ reduce $\mathbf{f}$ to Euclidean embeddings $\mathbf{h}_i = h_i(\mathbf{f})$, with $i \in \{1, 2\}$, $d_1 = 128$ and $d_2 = 2$. Subsequently, given a curvature scaler $c \in \mathbb{R}$, we apply the exponential map $\exp^c$ to each projection to obtain a hyperbolic representation in the Poincaré ball $\mathbb{D}_c^{d_i}$.

$$\mathbf{z}_1 = \exp^c(\mathbf{h}_1) \in \mathbb{D}_c^{128} \qquad \mathbf{z}_2 = \exp^c(\mathbf{h}_2) \in \mathbb{D}_c^2 \tag{5}$$

The high-dimensional branch captures fine-grained, class- and domain-specific features, whereas the low-dimensional branch introduces an information bottleneck, encouraging domain-invariant features and facilitating direct visualization of the embeddings (*cf.* Appendix A.1).

### 3.2.1 Domain-invariant cross-branch consistency

Each hyperbolic embedding $\mathbf{z}_i$ is then passed through a Multiclass Logistic Regression (MLR) head to produce logits $\hat{\mathbf{y}}_i = \text{MLR}(\mathbf{z}_i)$, where class scores are computed from geodesic distances in the Poincaré ball, as defined in Equations 2–3. The resulting logit vectors have a dimension equal to the number of classes for both branches $i = 1, 2$. During training, we apply the cross-entropy loss to both branches, but at test-time only the high-dimensional branch is used for inference. To promote domain-agnostic knowledge from the bottleneck branch into the main branch, we introduce a cross-branch consistency loss:

$$\mathcal{L}_{\text{KL}}(\hat{\mathbf{y}}_1, \hat{\mathbf{y}}_2) = T^2 \cdot \text{KL}\left(\sigma(\hat{\mathbf{y}}_2; T)\|\sigma(\hat{\mathbf{y}}_1; T)\right) \tag{6}$$

where KL is the Kullback–Leibler divergence, and $T \in \mathbb{R}$ is a temperature scaler. Notably, this cross-branch regularization works at the class-logit level, allowing information transfer between branches despite their differing embedding dimensions.

### 3.2.2 Overall objective

The final training objective combines the cross-entropy losses from both branches and the consistency loss, weighted with $\lambda \in \mathbb{R}$:

$$\mathcal{L} = \mathcal{L}_{\text{CE}}(\hat{\mathbf{y}}_1, \mathbf{y}) + \mathcal{L}_{\text{CE}}(\hat{\mathbf{y}}_2, \mathbf{y}) + \lambda \mathcal{L}_{\text{KL}}(\hat{\mathbf{y}}_1, \hat{\mathbf{y}}_2) \tag{7}$$

By merging domain-sensitive high-dimensional features $\mathbf{z}_1$ with a low-dimensional, domain-agnostic representation $\mathbf{z}_2$, our method learns both global, domain-invariant features and local, domain-specific features, implicitly taking advantage of the structure of the hyperbolic manifold. During inference, we only use the logits from the high-dimensional branch $\hat{\mathbf{y}}_1$, as they capture more fine-grained details, thereby providing richer semantic information.

## 4 Experimental results

In this section, we empirically evaluate and compare the advantages of representing medical image data within a hyperbolic manifold compared to the traditional Euclidean one. Our experiments are designed to assess three key aspects: (1) in-distribution (ID) image classification, (2) robustness against distribution shifts, and (3) the influence of manifold geometry and latent bottleneck size on robustness.

Across our experiments, we utilize a frozen pre-trained foundation model as a feature extractor, followed by a linear layer of dimension $d_1 = 128$ (Ermolov et al., 2022). This head is instantiated in *three variants* that form the basis of our evaluation. First, a Euclidean ERM is implemented using a standard Euclidean classification head with a softmax-based cross-entropy objective. Second, a hyperbolic counterpart (HypERM) projects the linear layer onto the Poincaré ball and applies Multiclass Logistic Regression (MLR) (Ganea et al., 2018). Third, our proposed cross-branch consistency approach (HypCBC) introduces an additional low-dimensional branch ($d_2 = 2$) and a KL loss controlled by temperature $T = 3$ and weight $\lambda = 0.2$. For hyperbolic models, we fix the curvature at $c = 1.0$, a commonly used default value (Mettes et al., 2024). This isolates the effect of hyperbolic geometry while avoiding an additional dataset-specific hyperparameter that could distort comparisons.

Furthermore, we employ a feature-clipping radius of $r = 1.0$ to ensure numerical stability (Guo et al., 2022). All models are trained using the AdamW optimizer (Loshchilov & Hutter, 2019) with cross-entropy loss and a cosine-annealing learning-rate schedule. We use an initial learning rate of $1 \times 10^{-4}$ with a batch size of 64, and apply early stopping after 10 epochs without improvement. The sensitivity to the hyperparameters of HypCBC is reported in Table 4.

### 4.1 Medical image classification

We first evaluate the accuracy performance achieved across eleven real-world medical datasets from the dataset collection of Yang et al. (2023), with default train–val–test splits and resolution 224×224. These datasets, detailed in Table 1, span a diverse range of imaging modalities (9), sample sizes ($10^2-10^5$), and label granularities ($2-11$). Notably, we exclude *Chest* from our experiments because of its multi-label setup.

We further evaluate three distinct ViT-based models: ViT-S (Dosovitskiy et al., 2021), pre-trained on ImageNet-21k following the training recipe of Steiner et al. (2022), DeiT3-S (Touvron et al., 2022), and DINOv2-S (Oquab et al., 2024). This allows us to assess whether the hyperbolic gains consistently transfer across models with different pre-training strategies. For each model and dataset, we report the average classification accuracy (over five seed runs) for both single-branch Euclidean (ERM) and hyperbolic classifiers (HypERM).

### 4.1.1 Results

The results reported in Table 2 demonstrate that, overall, hyperbolic representation learning consistently improves classification performance, although the magnitude of the gains varies depending on the specific dataset and model.

With ViT, hyperbolic embeddings outperform Euclidean on 8 of 11 tasks, with the largest gains on OCT (+1.32%), Tissue (+1.18%), and Pneumonia (+1.12%), a minor drop on Retina, and negligible changes on Blood and OrganA. With DeiT3, hyperbolic models score first on 10 of 11 datasets, most notably on Pneumonia, Derma, and Breast (gains from +1.41% to +1.89%). With DINOv2, the best model overall, hyperbolic embeddings outperform Euclidean ones in 10 out of 11 tasks, with the biggest increases on OCT (+2.70%), Path (+1.48%), and Tissue (+1.36%). To summarize, across all models and datasets, hyperbolic classifiers offer statistically significant gains over Euclidean ones (Wilcoxon signed-rank test, $p < 0.05$).

Table 1: Dataset details including data source, imaging modality, type of classification task (with number of classes), and predefined data splits. ML: Multi-Label, MC: Multi-Class, BC: Binary-Class, OR: Ordinary Regression. Table adapted from Doerrich et al. (2025).

| Dataset | Source | Imaging Modality | Task (# Classes) | Number of Samples Train / Val / Test |
|---|---|---|---|---|
| Blood | Acevedo et al. (2020) | Blood Cell Microscope | MC (8) | 11,959 / 1,712 / 3,421 |
| Breast | Al-Dhabyani et al. (2020) | Breast Ultrasound | BC (2) | 546 / 78 / 156 |
| Chest | Wang et al. (2017) | Chest X-Ray | ML-BC (2) | 78,468 / 11,219 / 22,433 |
| Derma | Tschandl et al. (2018) Codella et al. (2018) | Dermatoscope | MC (7) | 7,007 / 1,003 / 2,005 |
| OCT | Kermany et al. (2018) | Retinal OCT | MC (4) | 97,477 / 10,832 / 1,000 |
| OrganA | Bilic et al. (2023) Xu et al. (2019) | Abdominal CT | MC (11) | 34,561 / 6,491 / 17,778 |
| OrganC | Bilic et al. (2023) Xu et al. (2019) | Abdominal CT | MC (11) | 12,975 / 2,392 / 8,216 |
| OrganS | Bilic et al. (2023) Xu et al. (2019) | Abdominal CT | MC (11) | 13,932 / 2,452 / 8,827 |
| Path | Kather et al. (2019) | Colon Pathology | MC (11) | 89,996 / 10,004 / 7,180 |
| Pneumonia | Kermany et al. (2018) | Chest X-Ray | BC (2) | 4,708 / 524 / 624 |
| Retina | Liu et al. (2022a) | Fundus Camera | OR (5) | 1,080 / 120 / 400 |
| Tissue | Ljosa et al. (2012) | Kidney Cortex Microscope | MC (8) | 165,466 / 23,640 / 47,280 |

Table 2: Accuracy (averaged over five runs) of Euclidean (ERM) and hyperbolic (HypERM) classifiers on eleven medical datasets, evaluated with three ViT-based models. We highlight in bold the **best manifold** across each model and dataset. Notably, the hyperbolic representation is significantly better than the Euclidean one across all experiments (Wilcoxon signed-rank test, $p < 0.05$).

| | Br | Pn | Re | De | Bl | OrC | OrS | OrA | Pa | Ti | OCT | Avg |
|---|---|---|---|---|---|---|---|---|---|---|---|---|
| **ViT** | | | | | | | | | | | | |
| ERM | 82.56 | 87.21 | **61.90** | 81.65 | **97.68** | 85.42 | 76.09 | **91.50** | 93.48 | 61.04 | 74.02 | 81.14 |
| HypERM | **83.08** | **88.33** | 61.45 | **82.19** | 97.61 | **85.72** | **77.07** | 91.45 | **93.54** | **62.22** | **75.34** | **81.64** |
| **DeiT3** | | | | | | | | | | | | |
| ERM | 82.56 | 88.59 | 59.15 | 77.90 | 96.17 | 83.33 | **72.18** | 89.44 | 90.78 | 59.34 | 78.54 | 79.82 |
| HypERM | **83.97** | **90.48** | **59.65** | **79.44** | **96.50** | **83.61** | 72.03 | **89.48** | **91.23** | **60.61** | **78.90** | **80.54** |
| **DINOv2** | | | | | | | | | | | | |
| ERM | **85.77** | 89.94 | 64.65 | 82.77 | 97.90 | 88.04 | 76.98 | 92.82 | 92.46 | 61.95 | 83.60 | 83.35 |
| HypERM | 85.38 | **90.38** | **65.40** | **83.59** | **97.91** | **88.32** | **77.24** | **92.88** | **93.94** | **63.31** | **86.30** | **84.06** |

## 4.2 Domain Generalization

### 4.2.1 Datasets

We evaluate domain generalization performance under out-of-distribution (OOD) conditions on three established medical dataset collections.

*Fitzpatrick17k* (Groh et al., 2021) is a dermatological dataset including 16,577 samples labeled with three disease classes and skin-tone information according to the Fitzpatrick scale. The Fitzpatrick skin-type scale categorizes human skin tones from Type I (very light) to Type VI (deeply pigmented), providing a standardized measure of skin pigmentation from light to dark. We aggregate the skin tone information in three groups (*i.e.*, domains), as in Daneshjou et al. (2022): {I–II, III–IV, V–VI}. Given the limited number of domains, we assess the performance on a leave-one-domain-out protocol (LODO).

*Camelyon17-WILDS* (Bandi et al., 2018; Koh et al., 2021) is a standard binary domain generalization benchmark for histopathology consisting of 422,394 images acquired from five hospitals. We follow the default splits, training on three hospitals, validating on one, and testing on one.

For *Retina*, a 5-class diabetic retinopathy classification task, we construct cross-dataset shifts using four widely adopted fundus imaging datasets (Che et al., 2023; Zhou et al., 2023). APTOS 2019 (Karthik & Dane, 2019) and DeepDR (Liu et al., 2022b) serve as the in-distribution training data, comprising a total of 4,608 labeled samples. IDRiD (Porwal et al., 2018) forms the validation domain with 1,744 samples, while Messidor-2 (Decencière et al., 2014) provides the test domain with 7,000 samples. These datasets differ in acquisition devices, grading protocols, and patient cohorts, thereby inducing realistic distribution shifts.

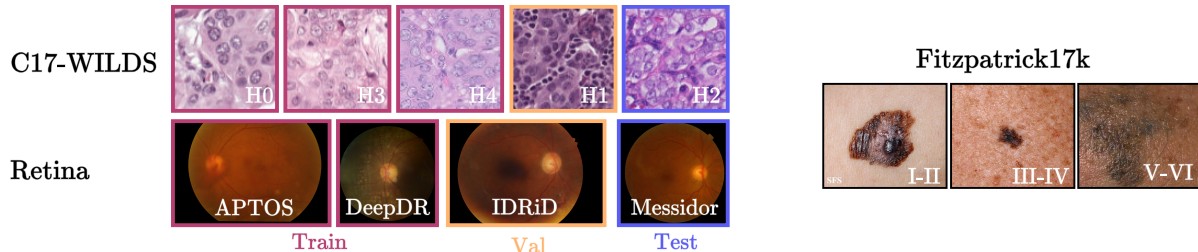

Figure 3: Overview of dataset domains. **Top-Left**: Camelyon17-WILDS, showing representative "tumor" patches from each contributing hospital (H0–H4). **Bottom-Left**: *Retina*, showing representative fundus images ($y = 4$) from each dataset domain (APTOS 2019, DeepDR, IDRiD, Messidor-2). Colored frames indicate the train (pink), validation (orange), and test (purple) subsets. **Right**: Fitzpatrick17k, showing representative "malignant" skin-lesion images from each of the three skin-tone groups (I–II, III–IV, V–VI).

### 4.2.2 Methods

We compare HypCBC against standard empirical risk minimization (ERM) and its hyperbolic variant (HypERM). In addition, we evaluate Euclidean embeddings enhanced with data augmentations such as RandAugment (Cubuk et al., 2020), AugMix (Hendrycks et al., 2019), and Med-C (Di Salvo et al., 2024). We also include established methods such as IRM (Arjovsky et al., 2019), GroupDRO (Sagawa et al., 2020), VREx (Krueger et al., 2021), DANN (Ganin et al., 2016), CDANN (Li et al., 2018c), CORAL (Sun & Saenko, 2016), MMD (Li et al., 2018b).

All methods use DINOv2 image embeddings and the default hyperparameters defined in DomainBed (Gulrajani & Lopez-Paz, 2021). We report the area under the receiver operating curve (AUC), averaged over five seeds to accommodate varying class imbalance and the generally more challenging (OOD) conditions.

Table 3: AUC averaged over five runs. Fitzpatrick17k is evaluated using a LODO strategy, reporting results for each fold, together with their macro-average. For Camelyon17-WILDS and Retina, both val and test set results are reported. We highlight in bold the **best two methods** and underline the **significantly best** (paired $t$-test, $p < 0.05$) on each dataset. Across all datasets, HypCBC is the significantly best method (Wilcoxon signed-rank test, $p < 0.05$).

| Method | F17k | | | | C17-WILDS | | Retina | | OOD |
|---|---|---|---|---|---|---|---|---|---|
| | I–II | III–IV | V–VI | Avg | Val | Test | Val | Test | **Avg** |
| ERM | 79.93±0.2 | 82.88±0.2 | 79.23±0.3 | 80.68 | 97.05±0.1 | 98.32±0.1 | 86.65±1.2 | 78.40±1.3 | 86.07 |
| Med-C | 80.15±0.1 | 82.80±0.4 | 79.66±0.2 | 80.87 | 97.14±0.1 | 98.22±0.0 | 84.85±0.2 | 77.13±0.5 | 85.71 |
| RandAug | 80.04±0.1 | 82.86±0.5 | 79.24±0.2 | 80.71 | 96.74±0.0 | 98.15±0.1 | **87.85±0.2** | 78.82±0.5 | 86.24 |
| AugMix | 80.26±0.1 | 82.86±0.5 | 79.25±0.2 | 80.79 | 96.50±0.1 | 98.06±0.1 | 86.73±0.3 | **79.80±0.4** | 86.21 |
| IRM | 76.57±0.7 | 80.20±0.4 | 78.84±0.4 | 78.54 | 96.98±0.1 | 98.17±0.2 | 85.04±0.4 | 75.64±0.5 | 84.49 |
| GroupDRO | 80.15±0.2 | 82.54±0.2 | 78.60±0.6 | 80.43 | 97.50±0.1 | 98.22±0.1 | 85.71±1.1 | 77.46±0.7 | 85.74 |
| VREx | 79.16±0.1 | 82.35±0.6 | 79.66±0.3 | 80.39 | 97.36±0.1 | **98.35±0.1** | 87.05±0.3 | 78.67±0.4 | 86.09 |
| DANN | 79.90±0.1 | 82.47±0.3 | 79.69±0.4 | 80.69 | 97.00±0.1 | 98.22±0.1 | 86.54±0.7 | 78.05±0.4 | 85.98 |
| CDANN | 79.71±0.1 | 82.44±0.5 | 80.02±0.4 | 80.72 | 97.04±0.0 | 98.23±0.1 | 85.76±0.9 | 76.92±1.0 | 85.73 |
| MMD | 78.82±0.1 | 82.15±0.4 | 79.56±0.3 | 80.18 | 97.02±0.2 | 98.27±0.1 | **87.42±0.2** | 79.07±0.4 | 86.05 |
| CORAL | 78.83±0.1 | 82.40±0.3 | 79.52±0.4 | 80.25 | 97.22±0.1 | 98.25±0.2 | 87.36±0.2 | 79.28±0.4 | 86.12 |
| HypERM | **81.93±0.1** | **84.31±1.0** | **83.56±0.3** | **83.27** | **97.95±0.6** | 98.07±0.2 | 87.23±0.9 | 79.39±0.7 | **87.49** |
| HypCBC | **82.34±0.3** | **86.28±0.2** | **84.27±0.3** | **84.30** | **98.04±0.3** | **98.33±0.3** | 87.34±0.7 | **80.48±0.5** | **88.15** |

### 4.2.3 Results

As shown in Table 3, Euclidean methods achieve comparable performance overall. Among augmentation methods, Med-C leads on Fitzpatrick17k, while RandAugment and AugMix lead on Retina. Representation-learning methods, such as VREx and MMD, also yield notable benefits on Camelyon17-WILDS and Retina. Regarding the hyperbolic manifold, HypERM (single-branch) outperforms its Euclidean counterpart in all settings except Camelyon17-WILDS (test), where our proposed method, HypCBC, exhibits comparable results with the top-scoring VREx. Notably, while VREx utilizes domain labels during training, our method does not require such information. On the Retina test split, HypCBC outperforms the best Euclidean method, AugMix, by 0.68%. Furthermore, hyperbolic methods exhibit the largest improvements on Fitzpatrick17k. Specifically, HypERM alone significantly boosts AUC, and HypCBC achieves additional significant gains (paired $t$-test, $p < 0.05$). This pattern aligns with the magnitude of the shift in skin tone groups. Indeed, Fitzpatrick17k presents the most extreme shift (*cf.* Figure 3), while the shifts observed in Camelyon17-WILDS and Retina are milder. Overall, HypERM delivers superior generalization across a broad range of real-world shifts, with an average improvement of +1.42% over Euclidean embeddings (ERM). Furthermore, HypCBC yields an additional statistically significant improvement of +0.66% (Wilcoxon signed-rank test, $p < 0.05$).

## 5 Ablation studies

### 5.1 Latent dimension and generalization

To quantify how the size and manifold of the low-dimensional branch govern the trade-off between domain invariance and label discrimination, we train *single-branch classifiers* (*i.e.*, no consistency constraint) with embedding dimension $d \in \{2, 16, 32, 64, 128\}$ in both Euclidean and hyperbolic spaces. We evaluate on three benchmarks: Fitzpatrick17k (three skin-tone groups, standard ID split), Camelyon17-WILDS (five hospitals), and the Retina cross-dataset (four sources). For Camelyon17-WILDS and Retina, we merge the original train/validation/test splits and re-partition the data into a 70/10/20 stratified split, thereby ensuring that all domain groups appear in train, validation, and test splits and enabling more reliable invariance estimates.

After training each model under identical hyperparameters as previous experiments, we freeze its projection head and fit two linear classifiers on the resulting embeddings: one to predict domain labels (domain-classification AUC, lower means more invariance) and one to predict disease labels (label-classification AUC, higher means more discriminative). Experiments with *non-linear* classifiers are available in Appendix A.3.

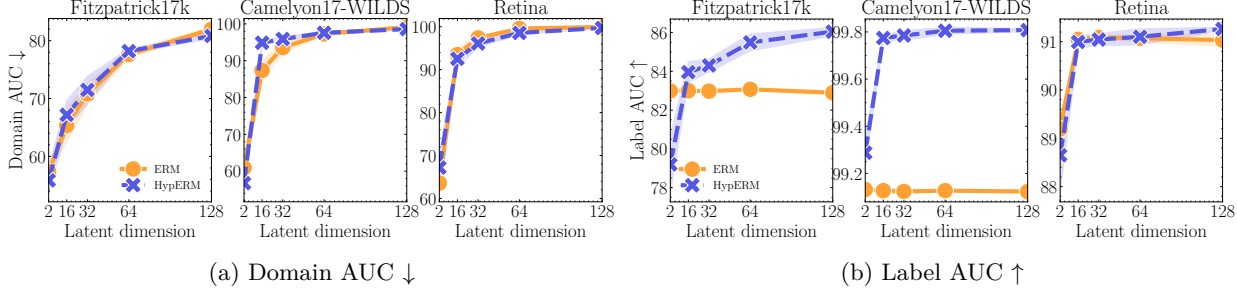

(a) Domain AUC ↓           (b) Label AUC ↑

Figure 4: Figure 4a shows the domain AUC (↓) vs. latent dimension $d$: lower is better (more domain-invariant), while Figure 4b plots the label AUC (↑) vs. $d$: higher is better (more discriminative). The curves are shown for Euclidean (ERM) and hyperbolic embeddings (HypERM). Results are reported on in-distribution splits where all domains appear in train/val/test, yielding $3, 5$, and $4$ domains for Fitzpatrick17k, Camelyon17-WILDS, and Retina, respectively.

### 5.1.1 Results

As shown in Figure 4a, the domain-classification AUC is lowest at $d = 2$ for both manifolds and steadily increases with $d$, confirming that a smaller bottleneck enforces stronger domain invariance. Crucially, Figure 4b demonstrates that even with a 2D bottleneck, label-classification AUC remains high (up to 99% on Camelyon17) and continues to improve with larger $d$. Notably, Euclidean embeddings plateau on Fitzpatrick17k and Camelyon17-WILDS, with no further increase on larger $d$. However, hyperbolic embeddings gain additional accuracy up to $d = 128$, especially on Fitzpatrick17k, with a gap of approximately 3% AUC. Taken together, these results suggest that (1) a very low-dimensional bottleneck provides strong domain invariance with only a modest loss in discriminative power, and (2) hyperbolic embeddings achieve higher label AUC than their Euclidean counterparts, with the largest gains observed at higher dimensions.

## 5.2 Hyperbolic cross-branch consistency

To isolate the contribution of our two-branch consistency regularization, we fix the high-dimensional branch $d_1 = 128$ and vary the low-dimensional bottleneck $d_2 \in \{2, 8, 16, 128\}$. For each configuration, we measure the AUC gain of the two-branch model over its single-branch counterpart on the same three benchmarks: Camelyon17-WILDS (test hospital), Retina (test dataset), and Fitzpatrick17k (leave-one-domain-out). This is evaluated on both hyperbolic and Euclidean manifolds.

### 5.2.1 Results

Figure 5 shows that in hyperbolic space, the largest improvements occur at $d_2 = 2$ and decline steadily as the bottleneck size grows, confirming that performance gains arise from compact regularization rather than added capacity. Indeed, $d_2 = 128$ slightly degrades hyperbolic performance. On Camelyon17, Euclidean cross-branch consistency (CBC) produces surprisingly negative AUC changes for all $d_2$, while hyperbolic CBC (HypCBC) yields positive gains at every reasonable bottleneck size. On Retina, both manifolds benefit at $d_2 = 2$. Although Euclidean gains appear larger at this point, the hyperbolic single-branch baseline itself is already 0.99% higher than Euclidean (*cf.* Table 3). On Fitzpatrick17k, HypCBC consistently improves performance across all folds, achieving up to a 2% AUC boost on the III-IV split versus only 0.5% for Euclidean. Overall, hyperbolic consistency regularization yields consistent, positive AUC gains across all datasets and bottleneck sizes, while Euclidean consistency produces uneven improvements.

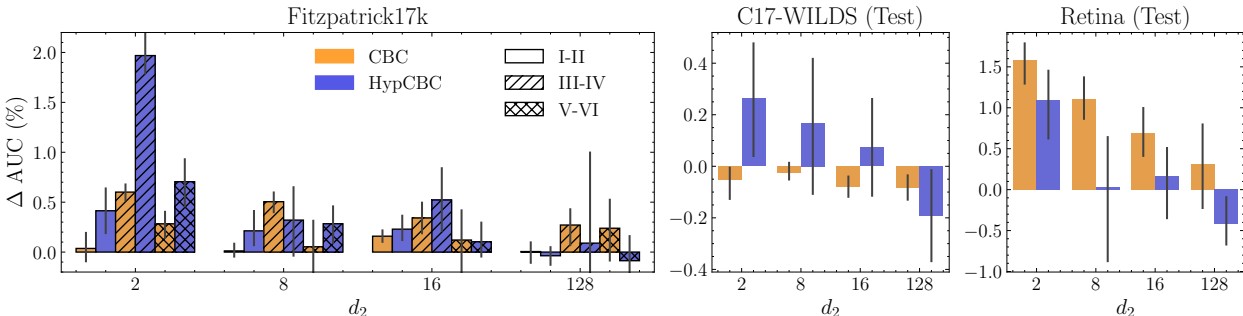

Figure 5: Improvement in AUC of two-branch consistency regularization over single-branch baseline ($\Delta$AUC) for Euclidean and hyperbolic manifolds, while varying bottleneck dimension $d_2$. These are termed CBC and HypCBC, respectively. From left to right: (1) Fitzpatrick17k leave-one-domain-out folds (I-II, III-IV, V-VI), differentiated with bar hatches, (2) Camelyon17-WILDS (test), and (3) Retina (test). While Euclidean regularization gains vary by dataset, hyperbolic regularization provides positive $\Delta$AUC across every task and reasonable (*i.e.*, $< 128$) bottleneck size.

## 5.3 Sensitivity analysis

Finally, we perform a sensitivity analysis over our two key hyperparameters: the KL weight $\lambda$, and the temperature $T$. We sweep over values for $\lambda \in \{0.1, 0.2, 0.5, 1.0\}$ and $T \in \{1.0, 3.0, 5.0, 10.0\}$ on Fitzpatrick17k (I–II, III–IV, V–VI), Retina (test), and Camelyon17-WILDS (test), reporting the mean AUC and its deviation across these settings (each averaged over five seeds).

### 5.3.1 Results

As indicated by the standard deviations reported in Table 4, performance varies only marginally with $\lambda$ and $T$ (*i.e.*, $\sigma \in [0.1, 0.7]$). This empirically demonstrates the robustness of our proposed method with respect to the chosen hyperparameters, thereby confirming its superiority against Euclidean baselines.

Table 4: Average AUC and standard deviation across different combinations of $\lambda$ and $T$, evaluated on Fitzpatrick17k, Camelyon17-WILDS, and Retina.

| F17k(I–II) | F17k(III–IV) | F17k(V–VI) | C17-WILDS | Retina |
|---|---|---|---|---|
| 81.96±0.4 | 85.55±0.7 | 83.81±0.4 | 98.30±0.1 | 80.19±0.2 |

## 6 Conclusion

### 6.1 Limitations

Our current implementation fixes the curvature parameter $c$ and relies on a frozen Euclidean backbone followed by lightweight hyperbolic MLR heads. While this design enables stable and efficient training, a static curvature may not be optimal across datasets or manifold dimensions. Allowing curvature to be learnable, or adopting data- or manifold-adaptive curvature schedules, could more effectively capture the geometric structure of each task. Moreover, although freezing the feature extractor offers substantial computational savings, fine-tuning the backbone or training a fully hyperbolic classifier may further amplify the benefits of the proposed approach, at the cost of increased training complexity. Finally, our method is evaluated only for single-label multi-class classification. Although the cross-branch consistency operates at the logit level and is conceptually extendable to multi-label settings, we do not explore this extension in this manuscript.

## 6.2 Discussion

Our work presents a hyperbolic representation learning framework for medical imaging that leverages the inherent hierarchical structure of clinical data. Across diverse modalities and scales, replacing Euclidean embeddings with our hyperbolic projections consistently improves in-distribution accuracy. Our two-branch, domain-invariant hyperbolic cross-branch consistency scheme further boosts out-of-distribution performance on three challenging benchmarks. Crucially, ablation studies confirm that these gains arise from the compact low-dimensional consistency regularization, not merely from added capacity, and that hyperbolic consistency outperforms its Euclidean counterpart. Overall, hyperbolic embeddings offer a straightforward yet powerful alternative for building robust, generalizable medical AI systems.

### Ethical and Data Quality Considerations

This work uses only publicly available datasets and adheres to their respective licenses. Fitzpatrick17k is known to contain substantial label noise ($\approx 22\%$) due to heterogeneous data sources and annotation procedures, as documented by recent analyses (Gröger et al., 2025). The dataset also presents limitations related to data provenance, as it was compiled from web-scraped images with limited transparency regarding original patient consent, and is distributed under a CC-BY-NC license. While these factors may affect absolute performance and raise broader ethical considerations, all methods in this study are evaluated under identical conditions, and Fitzpatrick17k remains a widely adopted dataset.

### Acknowledgements

This study was funded through the Hightech Agenda Bayern (HTA) of the Free State of Bavaria, Germany.

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

# A  Appendix

## A.1  Hyperbolic visualizations

Our method includes an explicit 2D hyperbolic branch, enabling direct visualization of learned representations without relying on post-hoc dimensionality reduction. Unlike techniques such as t-SNE or UMAP, which produce approximate low-dimensional embeddings and may distort neighborhood structure, our approach yields representations that are intrinsically two-dimensional and geometrically meaningful.

Figure 6 visualizes the learned 2D hyperbolic embeddings for Camelyon17-WILDS (train), Retina (train), and Fitzpatrick17k (train, fold V–VI). For each dataset, embeddings are shown in the Poincaré disk and colored by class (left) and by domain (right). Across all benchmarks, class structure is preserved, whereas domain labels exhibit substantial overlap and lack compact, locally separable clusters, suggesting reduced domain-specific organization.

On Camelyon17-WILDS, which comprises two classes and three hospital domains, a clear class separation is observed, whereas domain labels remain highly mixed. Notably, the training domains are imbalanced, with 131,696, 116,959, and 53,425 samples from hospital 4, 3, and 0, respectively. This explains the predominance of purple and orange points and the diffuse presence of the under-represented hospital 0 domain. Similar qualitative behavior is observed on Retina and Fitzpatrick17k. While Fitzpatrick17k shows a clearer class separation, Retina exhibits higher class overlap (especially with class 4). This is expected given the progressively graded nature of diabetic retinopathy severity, making class separation inherently more challenging.

To quantify these observations, we compute the average local $k$-NN entropy ($k = 15$) in hyperbolic space, measuring label diversity within local neighborhoods. Lower entropy indicates stronger local discrimination. On Camelyon17-WILDS, we observe a low class entropy ($\mathbb{H}_{\mathrm{class}} = 0.071$) and substantially higher domain entropy ($\mathbb{H}_{\mathrm{domain}} = 0.731$), confirming that neighborhoods are class-consistent but domain-mixed. This is facilitated by the large dataset size and its binary nature. Corresponding results for the remaining datasets are reported in Table 5. Higher class entropy on Fitzpatrick17k and Retina is expected due to increased task complexity, and for Fitzpatrick17k values are averaged across LODO splits, smoothing fold-specific effects. Overall, these results provide geometric evidence that the proposed representations preserve class structure while reducing domain-specific clustering.

Table 5: Average local $k$-NN entropy of 2D hyperbolic training embeddings, computed with respect to class labels ($\mathbb{H}_{\mathrm{class}}, \downarrow$) and domain labels ($\mathbb{H}_{\mathrm{domain}}, \uparrow$) on Fitzpatrick17k (3 classes, 2 train domains), Camelyon17-WILDS (2 classes, 3 train domains), and Retina (5 classes, 2 train domains). Lower class entropy indicates stronger class separability, while higher domain entropy reflects reduced domain-specific structure. For Fitzpatrick17k, entropy is averaged across the three training folds, while for Camelyon17-WILDS, results are computed on a subset of $30,000$ samples due to computational overhead. The results are consistent with the learned embeddings preserving class information while exhibiting limited domain discrimination.

| F17k | | C17-WILDS | | Retina | |
|---|---|---|---|---|---|
| $\mathbb{H}_{\mathrm{class}}(\downarrow)$ | $\mathbb{H}_{\mathrm{domain}}(\uparrow)$ | $\mathbb{H}_{\mathrm{class}}(\downarrow)$ | $\mathbb{H}_{\mathrm{domain}}(\uparrow)$ | $\mathbb{H}_{\mathrm{class}}(\downarrow)$ | $\mathbb{H}_{\mathrm{domain}}(\uparrow)$ |
| $0.414 \pm 0.04$ | $0.554 \pm 0.07$ | 0.071 | 0.731 | 0.259 | 0.605 |

## A.2  Standardized evaluation for Camelyon17-WILDS

Following the standard WILDS (Koh et al., 2021) evaluation protocol for Camelyon17-WILDS, we additionally report average accuracy alongside AUC (*cf.* Table 3) for both validation and test splits (*cf.* Table 6). The trends observed in AUC are consistent with those in accuracy. HypCBC achieves the best validation performance and competitive test accuracy within the standard deviation of the top-performing methods. Importantly, unlike GroupDRO and VREx, our method does not rely on domain labels during training.

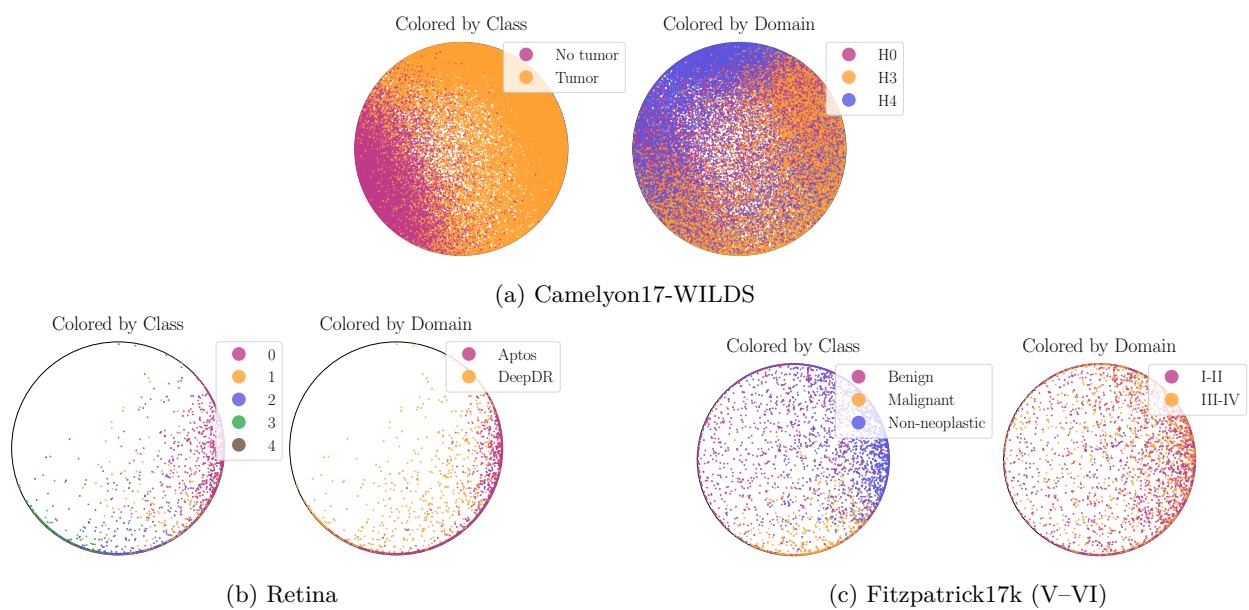

(a) Camelyon17-WILDS

(b) Retina

(c) Fitzpatrick17k (V–VI)

Figure 6: Visualization of 2D hyperbolic (training) embeddings learned by the proposed method, shown in the Poincaré disk. For each dataset, **left**: points colored by class label; **right**: the same embeddings colored by domain. While class separation is observed, domain separation is minimal.

Table 6: AUC and accuracy averaged over five runs for Camelyon17-WILDS val and test sets. We highlight in bold the **best two methods**.

| Method | AUC | | Accuracy | |
|---|---|---|---|---|
| | Val | Test | Val | Test |
| ERM | 97.05±0.1 | 98.32±0.1 | 91.23±0.2 | 94.02±0.2 |
| Med-C | 97.14±0.1 | 98.22±0.0 | 91.50±0.1 | 94.09±0.3 |
| RandAug | 96.74±0.0 | 98.15±0.1 | 90.48±0.1 | 93.89±0.6 |
| AugMix | 96.50±0.1 | 98.06±0.1 | 90.12±0.1 | 92.73±1.0 |
| IRM | 96.99±0.1 | 98.17±0.2 | 90.56±0.2 | 93.76±0.2 |
| GroupDRO | 97.50±0.1 | 98.22±0.1 | 91.79±0.2 | **94.30±0.2** |
| VREx | 97.36±0.1 | **98.35±0.1** | 91.49±0.3 | **94.39±0.3** |
| DANN | 97.00±0.1 | 98.22±0.1 | 91.07±0.2 | 93.53±0.4 |
| CDANN | 97.04±0.0 | 98.23±0.1 | 91.20±0.2 | 93.64±0.6 |
| MMD | 97.02±0.2 | 98.27±0.1 | 91.22±0.3 | 94.21±0.2 |
| CORAL | 97.22±0.1 | 98.25±0.2 | 91.32±0.3 | 94.04±0.2 |
| HypERM | **97.95±0.6** | 98.07±0.2 | **92.79±1.1** | 93.22±0.5 |
| HypCBC | **98.04±0.3** | **98.33±0.3** | **92.89±0.8** | 94.19±0.4 |

## A.3 Non-linear evaluation of domain invariance

Batch effects (*i.e.*, domain-specific acquisition or annotation artifacts) may be encoded nonlinearly (Rahman et al., 2023), in which case linear probes can underestimate the amount of domain information present in learned representations. To account for this, we replicate the domain-predictability experiment of Section 5.1, replacing the linear classifier with a two-layer MLP trained on frozen embeddings for 50 epochs using AdamW with a learning rate of $10^{-4}$.

As shown in Figure 7b, the nonlinear probe yields slightly higher absolute domain AUC than the linear classifier (Figure 7a), confirming that some domain information is indeed nonlinearly encoded. Crucially, the

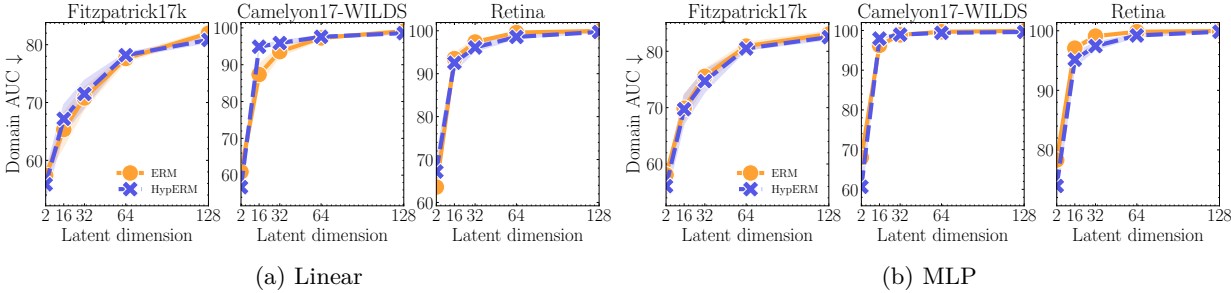

(a) Linear                                    (b) MLP

Figure 7: Figure 7a shows the domain AUC (↓) vs. latent dimension $d$ obtained with a linear classifier, while Figure 7b plots the domain AUC (↑) vs. $d$ obtained with an MLP. The curves are shown for Euclidean (ERM) and hyperbolic embeddings (HypERM). Results are reported on ID splits where all domains appear in train/val/test, yielding 3, 5, and 4 domains for Fitzpatrick17k, Camelyon17-WILDS, and Retina, respectively.

qualitative trends remain unchanged: domain predictability is minimized at low-dimensional bottlenecks, increases monotonically with embedding dimension, and exhibits consistent relative behavior across Euclidean (ERM) and hyperbolic (HypERM) manifolds.

We further observe dataset-dependent differences in the gap between linear and nonlinear probes. On Fitzpatrick17k and Camelyon17-WILDS, the difference in domain AUC between *linear* and *MLP* is marginal, suggesting that dominant domain confounders, such as skin tone and hospital-specific staining (*cf.* Figure 3), are largely texture-driven and close to linearly separable at higher dimensions. In contrast, the Retina benchmark exhibits a larger gap, likely due to more heterogeneous domain shifts arising from multiple acquisition devices and grading protocols, which induce more complex and less linearly separable domain structure. The absence of perfect domain separability on Fitzpatrick17k is also consistent with known ambiguity in skin-tone annotation.

Overall, while the degree of nonlinearity in domain information is dataset-dependent, the qualitative domain-invariance trends induced by low-dimensional bottlenecks remain consistent.

## A.4 Augmentations within the hyperbolic manifold

This section evaluates the interaction between HypCBC and common data augmentation strategies. We consider the same domain generalization benchmarks used throughout the paper, *i.e.*, Fitzpatrick17k, Camelyon17-WILDS, and Retina, and combine our methods with RandAugment and AugMix, which are among the strongest augmentation-based baselines in prior work.

Table 7 shows that the impact of data augmentation on HypCBC is dataset-dependent. While modest gains are observed on Fitzpatrick17k (up to +0.62%), no consistent improvements are seen on Camelyon17-WILDS or Retina. This indicates that the effectiveness of augmentations varies with dataset characteristics. Although the proposed bottleneck promotes domain-invariant representations, the model may still benefit from input diversity when such variability is informative.

Table 7: Average AUC for hyperbolic methods combined with RandAugment and AugMix across three domain generalization benchmarks, namely Fitzpatrick17k, Camelyon17-WILDS, and Retina.

| Method | F17k | | | | C17-WILDS | | Retina | |
| --- | --- | --- | --- | --- | --- | --- | --- | --- |
| | I–II | III–IV | V–VI | Avg | Val | Test | Val | Test |
| HypCBC | $82.34 \pm 0.3$ | $86.28 \pm 0.2$ | $84.27 \pm 0.3$ | 84.30 | $98.04 \pm 0.3$ | $98.33 \pm 0.3$ | $87.34 \pm 0.7$ | $80.48 \pm 0.5$ |
| w/AugMix | $82.53 \pm 0.1$ | $86.87 \pm 0.2$ | $85.37 \pm 0.3$ | 84.92 | $97.77 \pm 0.3$ | $98.52 \pm 0.1$ | $86.37 \pm 0.4$ | $80.08 \pm 0.5$ |
| w/RandAug | $82.43 \pm 0.1$ | $86.57 \pm 0.2$ | $84.83 \pm 0.3$ | 84.61 | $97.76 \pm 0.4$ | $98.52 \pm 0.1$ | $86.77 \pm 0.5$ | $79.83 \pm 0.4$ |

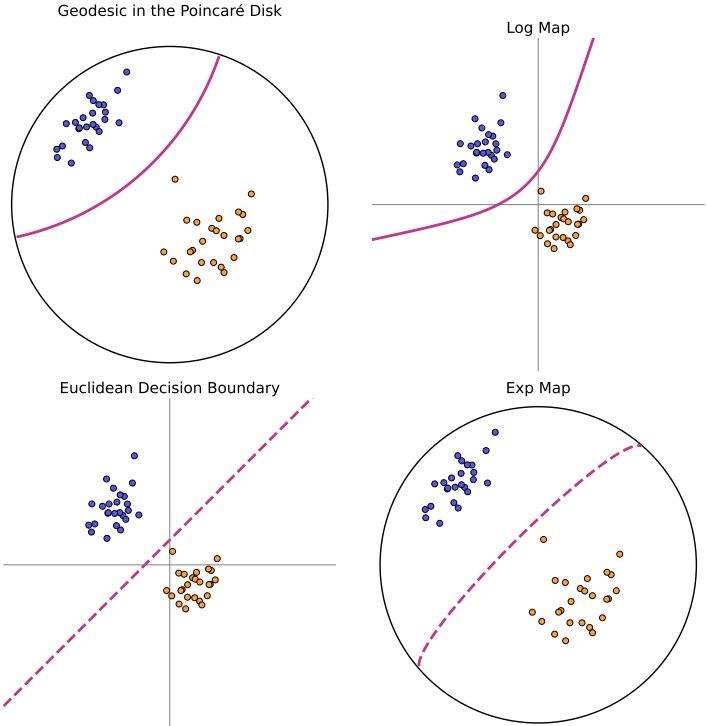

Figure 8: Toy illustration of Euclidean and hyperbolic decision boundaries under exponential and logarithmic maps, highlighting geometric differences between Euclidean linear classifiers and hyperbolic hyperplanes.

## A.5 Euclidean vs. Hyperbolic Decision Boundaries

To provide geometric intuition for the difference between Euclidean and hyperbolic classifiers, we include a two-dimensional toy example illustrating how decision boundaries behave under the exponential and logarithmic maps.

Figure 8 contrasts four corresponding views of the same classification setup. The top-left panel shows a hyperbolic decision boundary represented as a geodesic in the Poincaré disk, separating two synthetic clusters. In hyperbolic space, linear classifiers correspond to hyperbolic hyperplanes, which appear as circular arcs orthogonal to the unit disk. The top-right panel shows the same hyperbolic boundary after applying the logarithmic map to the Euclidean tangent space at the origin, where it becomes a non-linear curve. This illustrates how hyperbolic linearity translates into non-linear decision structure in Euclidean space.

For comparison, the bottom-left panel shows a standard Euclidean linear decision boundary in tangent space. The bottom-right panel visualizes this Euclidean boundary after mapping it into hyperbolic space via the exponential map. While the mapped boundary remains smooth, it does not correspond to a hyperbolic hyperplane, since hyperbolic decision boundaries are defined by geodesics in the manifold, whereas the exponential map of a Euclidean linear separator does not generally preserve geodesicity. This highlights the geometric mismatch between Euclidean linear classifiers and hyperbolic decision surfaces.

