# OpenReview forum: "HypCBC: Domain-Invariant Hyperbolic Cross-Branch Consistency for Generalizable Medical Image Analysis"
_TMLR — Accepted by TMLR_

### Review · Reviewer_RNcD · 2025-12-21

**Summary Of Contributions:**

The paper presents a study of hyperbolic embeddings in the domain of medical images. First, the background theory required for hyperbolic embeddings is reviewed. The paper then proposes to use pretrained vision transformers followed by a trainable linear layer and the hyperbolic exponential map to obtain good hyperbolic embeddings at a low cost. The embeddings are trained via a classification loss. Further, a dual branch “distillation” approach is proposed where a bottleneck low dimensional embedding is also learned in addition to a high dimensional embedding. The training proceeds via a classification loss for both the low and high dimensional embeddings as well as a KL-divergence loss between the logits produced by the low and high dimensional embeddings. The aim of adding the bottleneck low dimensional embedding is to obtain domain invariant features.

Experiments show first that hyperbolic embeddings outperform Euclidean ones and then that the proposed dual branch training scheme improves performance compared to only using hyperbolic embeddings. A broad set of ablations are carried out to evaluate the contributions of different components.

**Additional Comments:**

N/A

**Audience:**

Yes

**Audience Explanation:**

I believe that the findings are of interest to the medical image analysis community as well as the hyperbolic deep learning community.

**Claims And Evidence:**

Yes

**Claims Explanation:**

The main claims are supported by a series of experiments.

1. Hyperbolic embeddings are shown to outperform Euclidean embeddings.
2. The efficacy of the proposed 2-branch training strategy is demonstrated.
3. Ablations show the impact of varying the dimensionality of the latent dimensions.

**Requested Changes:**

### Critical

1. “Distillation” is not the best term to use for the proposed two-branch training strategy. Distillation typically refers to the training of a smaller network to mimic a frozen larger network. Unless there is prior work calling a similar strategy distillation, I recommend using a different term.
2. The experiments use three different ViTs. I recommend not to refer to these as different architectures, as they use minor variations of the same ViT architecture. Instead these could be differentiated as different “models” or different “training recipes”.

### Recommended for strengthening the paper

3. The fact that a 2D bottleneck proves to be useful opens up opportunities for visualizations. It would be very interesting to use visualizations of the 2D latents to see whether an interpretable hierarchical structure is obtained, as this is the stated goal of using hyperbolic embeddings.
4. The comparison between Euclidean embeddings and hyperbolic embeddings is interesting. It would be helpful for the reader if the difference between these was more explicitly described. The only two differences, as far as I understand, is the use of the exponential map for the hyperbolic embeddings and the use of hyperbolic hyperplanes for the classification rather than Euclidean hyperplanes. It should be possible to map the hyperbolic hyperplanes via the logarithmic map to the Euclidean tangent space, yielding curved decision surfaces in the Euclidean space. In the 2D case it would even be possible to visualize these 2D curves. Vice versa, one could visualize Euclidean decision surfaces (2D lines) after mapping them through the exponential map to obtain some curved decision surfaces in hyperbolic space, contrasting the hyperbolic hyperplanes in the Poincare disk model.

Visualizations as proposed in the two points above would help the reader understand the benefit provided by hyperbolic embeddings.

---

> ### Author Response · Authors · 2026-01-11
> **Answer RNcD**
>
> ## Regarding the term distillation
> > 1. “Distillation” is not the best term to use for the proposed two-branch training strategy. Distillation typically refers to the training of a smaller network to mimic a frozen larger network. Unless there is prior work calling a similar strategy distillation, I recommend using a different term.
>
> We thank the reviewer for this important clarification. While prior work has used the term distillation more broadly, including settings without a frozen teacher (Sultana et al., 2023), we agree that the term can be ambiguous in our context and may not best reflect the proposed training strategy.
>
> Following the reviewer’s suggestion, we have revised the terminology throughout the manuscript and now refer to our approach as cross-branch consistency (CBC). Accordingly, we updated the method name and title to “HypCBC: Domain-Invariant Hyperbolic Cross-Branch Consistency for Generalizable Medical Image Analysis.” This terminology more accurately captures the role of our auxiliary branch as a consistency-based regularizer rather than a classical distillation framework.
>
> Thank you.
>
> Reference:
> Sultana, Maryam, et al. "Self-distilled vision transformer for domain generalization." ACCV. 2022.
>
> ## Regarding the term backbone
>
> > 2. The experiments use three different ViTs. I recommend not to refer to these as different architectures, as they use minor variations of the same ViT architecture. Instead these could be differentiated as different “models” or different “training recipes”.
>
> We thank the reviewer for this clarification and acknowledge our imprecise use of the terms architecture and backbone. We have revised the manuscript accordingly.
>
> ## Regarding the visualization of the 2D embeddings
>
> > 3. The fact that a 2D bottleneck proves to be useful opens up opportunities for visualizations. It would be very interesting to use visualizations of the 2D latents to see whether an interpretable hierarchical structure is obtained, as this is the stated goal of using hyperbolic embeddings.
>
> We thank the reviewer for this suggestion and giving us the opportunity to strengthen our contribution. We have added Poincaré disk visualizations of the 2D hyperbolic embeddings for all datasets, coloring points by class and by domain (cf. Figure 6, Appendix A.1).  While explicit hierarchical structure can be more naturally visualized in fine-grained classification or retrieval settings (e.g., Ermolov et al., 2022; Wang et al., 2025), our visualizations qualitatively show that class structure is preserved while no clear domain separation can be observed.
>
> To complement the qualitative analysis, we also report a quantitative evaluation based on local k-NN entropy in hyperbolic space, measuring neighborhood entropy with respect to class and domain labels. Consistent with the visual evidence, we observe lower entropy for class labels and higher entropy for domain labels, indicating class-consistent but domain-mixed local neighborhoods. For Fitzpatrick17k, entropy values are aggregated across the three LODO splits, which smooths fold-specific effects.
>
> References:
>
> Ermolov, Aleksandr, et al. "Hyperbolic vision transformers: Combining improvements in metric learning." CVPR. 2022.
>
> Wang, Ziwei, et al. "Learning visual hierarchies in hyperbolic space for image retrieval." ICCV. 2025.
>
> ## Regarding the visualization of the decision boundaries
>
> > 4. The comparison between Euclidean embeddings and hyperbolic embeddings is interesting. It would be helpful for the reader if the difference between these was more explicitly described.
>
> We thank the reviewer for this insightful suggestion. To qualitatively address it, we added a two-dimensional toy example (Appendix A.5) that explicitly visualizes the relationship between Euclidean and hyperbolic decision boundaries using the exponential and logarithmic maps.
>
> As pointed out by the reviewer, the example shows that geodesic decision boundaries in the Poincarè disk, which become non-linear when mapped to Euclidean tangent space. Conversely, Euclidean linear decision boundaries, when mapped into hyperbolic space via the exponential map, do not generally correspond to hyperbolic hyperplanes.
>
> We believe that this qualitative visualization helps clarify the geometric interaction between Euclidean and hyperbolic manifolds, and provides intuitive insight into how hyperbolic classifiers encode decision boundaries differently from their Euclidean counterparts.

---

### Review · Reviewer_BQiW · 2025-12-29

**Summary Of Contributions:**

This paper addresses the fragility of medical AI under distribution shifts by proposing HypDIST, an unsupervised domain generalization framework. The authors argue that Euclidean spaces fail to capture the hierarchical nature of medical data and propose using the Poincaré ball model of hyperbolic geometry.\
Methodology: The core innovation is a dual-branch architecture where a high-dimensional branch ($d=128$) is distilled from a low-dimensional hyperbolic bottleneck ($d=2$). This bottleneck forces the model to discard "style" (domain) information while retaining "content" (labels).\
Key Strengths: Extensive validation across 11 modalities; strong geometric motivation; achieves domain invariance without requiring explicit domain labels.\
Key Weaknesses: Use of non-standard evaluation metrics for the WILDS benchmark and omission of very recent, highly relevant hyperbolic DG literature.

**Audience:**

Yes

**Audience Explanation:**

This work intersects three major interests within the TMLR community:

Medical AI: Practitioners seeking unsupervised ways to handle hardware and population shifts.

Geometric Deep Learning: Researchers interested in the "intrinsic dimension" of medical data and the practical utility of non-Euclidean manifolds.

AI Fairness: Those investigating technical pathways to mitigate demographic bias (e.g., skin tone) via representation learning.

**Broader Impact Concerns:**

"Fairness Washing": There is a risk that "domain invariance" is interpreted as a complete solution for "fairness." The authors should explicitly state that if training data is fundamentally biased (e.g., missing specific pathologies for certain skin tones), a robust model may still fail to represent those groups accurately.

Data Provenance: The paper relies on scraped data (Fitzpatrick17k). The authors should acknowledge the ethical limitations regarding patient consent in such datasets to promote responsible data curation practices in the community.

**Claims And Evidence:**

Yes

**Claims Explanation:**

The claims are supported by a rigorous benchmarking of 11 in-distribution datasets and 3 OOD benchmarks. The ablation studies (Figure 4) convincingly demonstrate that 2D hyperbolic space preserves label discriminability significantly better than 2D Euclidean space under extreme compression. However, the evidence is weakened by:

Metric Misalignment: For Camelyon17-Wilds, the authors report AUC, whereas the standard evaluation metric is Average/Worst-group Accuracy. This makes comparison with the official leaderboard difficult.

Baselines: The paper fails to compare against concurrent hyperbolic domain generalization methods (e.g., Bi et al., AAAI 2025), which limits the "State-of-the-Art" claim.

**Requested Changes:**

1. Metric Standardization: Update the Camelyon17-Wilds results to include Average Accuracy and Worst-Group Accuracy to align with the WILDS benchmark standards.
2. Ethical & Data Quality Statement: Add a section addressing the known issues with the Fitzpatrick17k dataset, including the ~22% label error rate and the CC-BY-NC licensing/provenance concerns.
3. Comparison with Concurrent SOTA: Discuss and, if possible, quantitatively compare against Bi et al. (AAAI 2025) and Yang et al. (IEEE TIP 2025). These are direct hyperbolic competitors that should be acknowledged to contextualize the novelty of HypDIST.
4. Embedding Visualization: Provide Poincaré disk visualizations of the 2D embeddings for Fitzpatrick17k. Color points by domain vs. class to qualitatively demonstrate the effectiveness of the domain-invariant bottleneck.
5. Technical Clarity: Explicitly state the pre-training data for the backbones (ImageNet-1k vs. 21k) and provide a brief justification for fixing curvature c=1.0.

---

> ### Author Response · Authors · 2026-01-11
> **Answer BQiW**
>
> ## Regarding Metric Standardization
> > 1. Metric Standardization: Update the Camelyon17-Wilds results to include Average Accuracy and Worst-Group Accuracy to align with the WILDS benchmark standards.
>
> We thank the reviewer for pointing this out. We have now extended the evaluation of Camelyon17-WILDS to include average accuracy in addition to AUC (cf. Appendix A.2, Table 6), in line with the standard WILDS reporting protocol.
>
> We note that worst-group accuracy is not reported for Camelyon17-WILDS by design. In the official benchmark, both the validation and test splits correspond to single held-out domains, rather than multiple simultaneously evaluated groups, which precludes the definition of a worst-group metric.
>
> The updated results show that the trends observed in AUC are consistent with those in accuracy, supporting the conclusions drawn in the main paper. In particular, HypCBC achieves the best validation accuracy and performs within the standard deviation of the top-performing method on the test set, despite being fully domain-agnostic.
>
> ## Regarding Ethical & Data Quality Statement
>
> > 2. Ethical & Data Quality Statement: Add a section addressing the known issues with the Fitzpatrick17k dataset, including the ~22% label error rate and the CC-BY-NC licensing/provenance concerns.
>
> We thank the reviewer for raising this important point. We have added an Ethical and Data Quality Considerations section explicitly acknowledging the known limitations of Fitzpatrick17k (cf. end of the manuscript).
>
> ## Regarding the comparison with concurrent SOTA
> > 3. Comparison with Concurrent SOTA: Discuss and, if possible, quantitatively compare against Bi et al. (AAAI 2025) and Yang et al. (IEEE TIP 2025). These are direct hyperbolic competitors that should be acknowledged to contextualize the novelty of HypDIST.
>
> We thank the reviewer for highlighting the importance of positioning our method with respect to concurrent hyperbolic approaches. We have accordingly strengthened the related work section (cf. Sections 2.2-2.3) to explicitly contrast our contribution with Bi et al. (AAAI 2025) and Yang et al. (IEEE TIP 2025).
>
> Bi et al. embed hyperbolic geometry directly into VMamba via end-to-end state-space modeling for fine-grained domain generalization. By contrast, our method is backbone-agnostic and introduces lightweight hyperbolic projections with cross-branch consistency on frozen features, targeting unsupervised domain generalization with minimal architectural overhead.
>
> Yang et al. introduce a hyperbolic knowledge distillation approach for cross-domain few-shot learning. Their method relies on multiple domain-specific teacher models, meta-learning, and access to target-domain data at test time, placing it closer to domain adaptation than domain generalization. In contrast, our setting assumes no domain labels and no access to target-domain data at any stage, and focuses on improving robustness to entirely unseen domains under a standard domain generalization protocol
>
> Due to these differing assumptions and evaluation protocols, and the absence of publicly available code for both methods at the time of writing, a direct quantitative comparison is not applicable.
>
> ## Regarding embedding visualization
> > 4. Embedding Visualization: Provide Poincaré disk visualizations of the 2D embeddings for Fitzpatrick17k. Color points by domain vs. class to qualitatively demonstrate the effectiveness of the domain-invariant bottleneck.
>
> We thank the reviewer for this suggestion and for giving us the chance to strengthen our contribution. We have added Poincaré disk visualizations of the 2D hyperbolic embeddings for all datasets, coloring points by class and by domain. These visualizations qualitatively show that class structure is preserved while no clear domain separation can be observed (cf. Figure 6, Appendix A.1).
>
> To complement the qualitative analysis, we also report a quantitative evaluation based on local k-NN entropy in hyperbolic space, measuring neighborhood entropy with respect to class and domain labels. Consistent with the visual evidence, we observe lower entropy for class labels and higher entropy for domain labels, indicating class-consistent but domain-mixed local neighborhoods.
>
> ## Regarding technical clarity
> > 5. Technical Clarity: Explicitly state the pre-training data for the backbones (ImageNet-1k vs. 21k) and provide a brief justification for fixing curvature c=1.0.
>
> We thank the reviewer for pointing this out. We have now explicitly clarified the pre-training setup of all backbones (ImageNet-21k) in Section 4.1.
>
> In addition, we have expanded the justification for fixing the curvature parameter. Specifically, for hyperbolic models we fix the curvature to c = 1.0, a commonly adopted default in prior hyperbolic work. This choice isolates the effect of hyperbolic geometry itself and avoids introducing an additional dataset-specific hyperparameter that could confound comparisons across datasets and models.

---

### Review · Reviewer_5Qjm · 2026-01-02

**Summary Of Contributions:**

This paper proposes a hyperbolic representation learning framework for medical image analysis. The authors introduce an unsupervised two-branch hyperbolic distillation strategy that promotes domain-invariant features via a low-dimensional bottleneck. The authors demonstrated that hyperbolic embeddings consistently outperform Euclidean embeddings on Med-MNIST across three Vision Transformer backbones.
On three domain-generalization benchmarks, the author demonstrated that, in most cases, the proposed technique provides better results.

**Audience:**

Yes

**Audience Explanation:**

Out-of-domain generalization remains a major challenge in medical image analysis. At present, the authors have not sufficiently explained or demonstrated the effectiveness of their technique. However, once these issues are addressed, it would be of considerable interest to the TMLR audience.

**Broader Impact Concerns:**

No direct ethical concern is apparent to me.

**Claims And Evidence:**

No

**Claims Explanation:**

Even though the technique shows better performance, it is not clear if these gains are due to "out-of-domain" robustness.

To be specific
- The authors did not quantify that the boost in performance is due to the 'domain invariant' nature of  distillation

The empirical way to demonstrate is to compare the IN-domain and out-of-domain performance of different models across different baselines and quantify the drop in performance, i.e.,

$\Delta AUC = AUC_{IN} - ACC_{OUT}$

Also, the authors did not present either a theoretical or an intuitive explanation for why the proposed distillation would be domain-invariant.

**Requested Changes:**

1. Demonstrate the domain invariant nature of distillation. Please compare the IN-domain and out-of-domain performance of different models across different baselines and quantify the drop in performance.


2. " However, end-to-end hyperbolic networks can be unstable on large datasets (Ayubcha et al., 2024)" -- I could not find in the reference where the instability in end-to-end training of hyperbolic networks is discussed. Please make it clear.

3. I am not sure if equations 2 and 3 are used in the technique. If not, they may be removed

4. It is not clear what type of data augmentations were done during training. The proposed distillation appears to be data augmentation-agnostic. How would the model perform if the augmentation bases, such as RandAug/Cutmix, are combined with the proposed distillation? if this is a bad idea, then please explain the reason.

5. " , we freeze its projection head and fit two linear classifiers on the resulting embeddings: one to predict domain labels (domain-classification AUC, lower means more invariance) and one to predict disease labels (label-classification AUC, higher means more discriminative)." - This assumes that the batch effect can be identified via linear models. However, batch effects can be nonlinear [ see a]. In that case, it cannot be well predicted by the linear regressors.

6. "... with a 2D bottleneck, label-classification AUC remains high (up to 99% on Camelyon17) and continues to improve with larger d." -- This implies that the task is relatively easy. In fact, for both "Fitzpatrick17k" and "Camelyon17-Wilds", the performance remained pretty high even for a 2-dimensional latent space. And only for "Retina" do we see improvements when increasing the latent size (a difficult task, in a sense, as it requires a larger representation space for classification). And from Table 3 e can see the pattern that the propose technque is performing well for the classification tasks that inhirently require smaller latent vecctor. And performing worse compared to the baseline on Ratina, which requires a larger embedding. Please add an explanation for this. If the observation is true, then add more competitive baselines.

7. The proposed technique has a limitation that it can not be applied to multi-label classification. I am not if it can be extended to segmentation. Please add clarification on this.






a. BEENE: deep learning-based nonlinear embedding improves batch effect estimation

---

> ### Author Response · Authors · 2026-01-11
> **Answer 5Qjm part 1**
>
> ## Regarding AUC drop
> > 1. Demonstrate the domain invariant nature of distillation. Please compare the IN-domain and out-of-domain performance of different models across different baselines and quantify the drop in performance.
>
> When using training performance as the in-distribution reference, our method exhibits a comparable AUC drop than competing approaches. However, we respectfully argue that a comparable or larger AUC drop does not necessarily indicate weaker robustness. The training splits already exhibit substantial internal heterogeneity in patient demographics, acquisition conditions, and devices. Moreover, differences between ID and OOD performance can arise from multiple factors beyond the targeted domain shift itself, including variations in disease severity and stage, which are not explicitly controlled in these benchmarks.
>
> As a result, methods that learn more robust and generalizable representations are also expected to achieve stronger in-distribution performance, which naturally leaves more room for absolute or relative degradation under distribution shift.
>
> Importantly, widely adopted domain generalization benchmarks and evaluation protocols, such as WILDS (Koh et al., 2021) or DomainBed (Gulrajani et al., 2021), emphasize absolute performance rather than relative performance drops. Under this evaluation setting, our method consistently achieves superior or competitive OOD performance without relying on domain information, supporting improved generalization. For this reason, we consider absolute OOD performance a more faithful indicator of robustness in our context.
>
> References:
>
> Koh, Pang Wei, et al. "Wilds: A benchmark of in-the-wild distribution shifts." ICML. 2021.
>
> Gulrajani, Ishaan, and David Lopez-Paz. "In Search of Lost Domain Generalization." ICLR. 2021.
>
> ## Regarding reference works
> > 2. " However, end-to-end hyperbolic networks can be unstable on large datasets (Ayubcha et al., 2024)" -- I could not find in the reference where the instability in end-to-end training of hyperbolic networks is discussed. Please make it clear.
>
> We thank the reviewer for pointing this out and agree that our original wording required clarification.
>
> Ayubcha et al., (2024) report in the abstract, “HCNNs (Hyperbolic CNNs) encounter efficiency and performance challenges with larger, complex datasets.” In the discussion, they further clarify that “we observed significant numerical instability with these methods” when convolutional operations are moved fully into hyperbolic space (Van Spengler et al., 2023; Bdeir et al., 2023).
>
> While the authors emphasize that “we did not observe numerical instability in the hybrid models,” they specify that “the computational efficiency of HCNNs was observed to be dramatically lower, with convergence requiring three to four times more epochs with similar hyperparameters,” and that “this was even more pronounced in larger datasets, limiting our ability to use larger datasets.”
>
> To report these findings more accurately, we revised our statement to emphasize convergence and scalability rather than implying universal numerical failure. The manuscript now reads:
>
> “However, end-to-end hyperbolic networks can exhibit numerical instability and substantially reduced training efficiency, with convergence requiring significantly more epochs (Ayubcha et al., 2024). This effect becomes more pronounced on larger datasets (Ayubcha et al., 2024).”
>
> ## Regarding equations
> > 3. I am not sure if equations 2 and 3 are used in the technique. If not, they may be removed
>
> Equations 2 and 3 are both operational in our model. Specifically, Equation 3 defines the hyperbolic geodesic distance used by the Multiclass Logistic Regression (MLR) classifier to compute class logits in the Poincaré ball. Equation 2 defines the Möbius addition required for these geodesic distance computations. We have clarified this dependency explicitly in the manuscript (cf. Section 3.2.1).

---

> > ### Comment · Reviewer_5Qjm · 2026-01-22
> >
> > ## Regarding AUC drop
> >  - `A comparable or larger AUC drop does not necessarily indicate weaker robustness.` - Why is this true? A model with perfectly robust generalization should perform the same on data from different sources.
> > - `OOD performance can arise from multiple factors beyond the targeted domain shift itself, including variations in disease severity and stage.` -- This technique does not require domain information. It is not clear whether any `targeted domain shift itself` is considered here.

---

> > > ### Author Response · Authors · 2026-01-24
> > > **Answer to Reviewer 5Qjm**
> > >
> > > ## Regarding AUC drop
> > >
> > > We thank the reviewer for the continued discussion and are pleased that the remaining concerns appear to be satisfactorily resolved.
> > >
> > > Regarding AUC drop, our position is that it is not a reliable proxy for robustness when interpreted in isolation. In-distribution training data generally exhibits unannotated heterogeneity (e.g., patient demographics, disease severity, acquisition protocols), which cannot explicitly be measured or controlled in most real-world settings. As a result, methods that learn more robust and generalizable representations are also expected to achieve stronger in-distribution performance.
> > >
> > > Conversely, the out-of-distribution splits isolate a single known confounder (e.g., skin tone), while other factors that influence performance remain uncontrolled. For this reason, we argue that it is neither realistic nor meaningful to expect perfectly invariant generalization across data sources in these real-world settings.
> > >
> > > We reiterate that the experimental setup and evaluation protocol follow established practice in domain generalization benchmarks such as WILDS (Koh et al., 2021) and DomainBed (Gulrajani et al., 2021) and are consistent with prior work in the literature. Our extensive experimental evaluation, including comprehensive ablation studies, strongly supports the scope and validity of our claims. This is also acknowledged explicitly by reviewers BQiW and RNcD.
> > >
> > > Lastly, we clarify that by “targeted domain shift” we refer to the benchmark-defined evaluation split, not to any information used by the model. Our method is fully domain-agnostic and does not rely on domain labels or explicit knowledge of the shift.
> > >
> > > References:
> > > Koh, Pang Wei, et al. "Wilds: A benchmark of in-the-wild distribution shifts." International conference on machine learning. PMLR, 2021.
> > >
> > > Gulrajani, Ishaan, and David Lopez-Paz. "In Search of Lost Domain Generalization." ICLR. 2021.

---

> ### Author Response · Authors · 2026-01-11
> **Answer 5Qjm part 2**
>
> ## Regarding data augmentation
> > 4. It is not clear what type of data augmentations were done during training. The proposed distillation appears to be data augmentation-agnostic. How would the model perform if the augmentation bases, such as RandAug/Cutmix, are combined with the proposed distillation? if this is a bad idea, then please explain the reason.
>
> We thank the reviewer for this important question. In the main experiments, unless otherwise stated, models are trained without data augmentation to isolate representation-level effects. To evaluate whether the proposed cross-branch consistency interacts negatively with input-level augmentations, we combine HypCBC with RandAugment and AugMix on Fitzpatrick17k, Camelyon17-Wilds, and Retina (Appendix A.4).
>
> Empirically, the impact of data augmentation on HypCBC is dataset-dependent. While modest gains are observed on Fitzpatrick17k (up to +0.62%), no consistent improvements are seen on Camelyon17-WILDS or Retina. This indicates that the effectiveness of augmentations varies with dataset characteristics. Although the proposed bottleneck promotes domain-invariant representations, the model may still benefit from input diversity when such variability is informative.
>
> ## Regarding batch effects
> > 5. This assumes that the batch effect can be identified via linear models. However, batch effects can be nonlinear [ see a]. In that case, it cannot be well predicted by the linear regressors.
>
> We thank the reviewer for the insightful remark, which allows us to strengthen our analysis of domain invariance. As noted, batch effects may be encoded nonlinearly. To address this, we extended our linear analysis by replicating the domain-predictability experiment using a two-layer MLP trained on frozen embeddings (50 epochs, AdamW, lr=$10^{-4}$).
>
> As expected, the nonlinear probe yields just slightly higher absolute domain AUC than the linear classifier, confirming that some domain information is indeed nonlinearly encoded. Importantly, however, the qualitative trends remain unchanged: domain predictability is minimized at low-dimensional bottlenecks, increases monotonically with embedding dimension, and exhibits consistent relative behavior across Euclidean (ERM) and hyperbolic (HypERM) representations. A more detailed analysis, including dataset-specific differences, is provided in Appendix A.3.
>
> ## Regarding 2D performance
> > 6. From Table 3 we can see the pattern that the propose technque is performing well for the classification tasks that inhirently require smaller latent vecctor. And performing worse compared to the baseline on Ratina, which requires a larger embedding
>
> We agree that the high performance observed with a 2D bottleneck on Fitzpatrick17k and Camelyon17-Wilds can be task-dependent. Both datasets are largely texture- and color-driven, where disease patterns can be encoded by compact representations, leading to high AUC even at low dimensionality. In contrast, Retina requires fine-grained structures, which may benefit from higher-capacity embeddings, explaining the larger gains with increasing dimension.
>
> However, we clarify that the results shown in Figure 2(b) are obtained under the evaluation protocol described in Section 5.1: for Camelyon17-WILDS and Retina, we merge the original train/val/test splits and apply a stratified split. This setup was adopted to increase the number of domain groups for domain-predictability analysis and to ensure a fair comparison of label-classification performance across embedding dimensions. Therefore, these results are not directly comparable to the ones observed in Table 3.
>
> ## Regarding multi-label and segmentation problems
> > 7. The proposed technique has a limitation that it can not be applied to multi-label classification. I am not if it can be extended to segmentation. Please add clarification on this.
>
> Our cross-branch consistency operates at the level of class logits and therefore is applicable for multi-label problems. This requires replacing the softmax cross-entropy with a sigmoid-based multi-label objective and applying the consistency constraint on a per-label basis. We did not evaluate this extension in the present work and now explicitly note this as a limitation and a direction for future research.
>
> Regarding segmentation, although hyperbolic representations have been studied for dense prediction, adapting our framework would require nontrivial modifications (e.g., pixel- or region-wise consistency objectives) and is therefore beyond the scope of this submission.

---

### Author Response · Authors · 2026-01-11
**General statement**

We thank the reviewers for their constructive feedback. We are encouraged that multiple reviewers recognized the strengths of our work, including the extensive validation across 11 established medical datasets and 3 domain-generalization benchmarks, the strong geometric motivation, and the ability to promote domain-invariant representations without requiring domain labels (BQiW, RNcD). Reviewers also highlighted that our main claims are supported by rigorous benchmarking and comprehensive ablation studies (BQiW, RNcD), and that the analysis convincingly demonstrates the advantage of low-dimensional hyperbolic embeddings over Euclidean ones under extreme compression (BQiW). We also appreciate the acknowledgment that hyperbolic embeddings consistently outperform Euclidean ones, and that our method yields improved performance on established benchmarks (5Qjm).
We are pleased that the reviewers identified the relevance of our findings to multiple communities, including medical image analysis, geometric deep learning, and fairness-oriented representation learning (BQiW, RNcD).

We have carefully revised and strengthened the manuscript throughout, enhancing clarity and presentation, highlighting relevant changes in blue. In particular, the Appendix now includes: (i) visualizations of the low-dimensional hyperbolic branch (Section A.1), (ii) standardized accuracy results for Camelyon17-WILDS (Section A.2), (iii) a non-linear evaluation of domain invariance (Section A.3), (iv) an analysis of data augmentation within the hyperbolic manifold (Section A.4), and (v) a qualitative comparison between Euclidean and hyperbolic decision boundaries (Section A.5).

Finally, to better reflect the nature of our approach and address concerns regarding terminology, we have updated the paper title to “HypCBC: Domain-Invariant Hyperbolic Cross-Branch Consistency for Generalizable Medical Image Analysis”. We will update the title on OpenReview in the final version.

We address all individual points below and welcome the opportunity for continued discussion to further improve the manuscript.

---

### Decision · Action_Editor_derZ · 2026-01-29

**Recommendation:** Accept as is

**Additional Comments:**

In this paper, the authors learn hyperbolic representations for medical images. All reviewers agree that the claims made in the paper are backed by evidence, and that the results will be of interest to those working in medical image analysis. I thus recommend acceptance.

**Audience:**

Yes

**Audience Explanation:**

Yes, reviewers unanimously agree.

**Claims And Evidence:**

Yes

**Claims Explanation:**

Yes, reviewers unanimously agree.

---

> ### Author Response · Authors · 2026-02-03
> **Response to Action Editor**
>
> The authors sincerely thank the Action Editor and reviewers for their time and thoughtful feedback, which helped to improve the quality of our work. We have submitted the camera-ready version, incorporating the suggestions provided during the review process.